# Serological Survey of Small Ruminant Lentivirus Infections in Free-Ranging Mouflon and Chamois in Slovenia

**DOI:** 10.3390/ani12081032

**Published:** 2022-04-15

**Authors:** Urška Kuhar, Diana Žele Vengušt, Gorazd Vengušt

**Affiliations:** 1Veterinary Faculty, Institute of Microbiology and Parasitology, University of Ljubljana, Gerbičeva 60, 1000 Ljubljana, Slovenia; urska.kuhar@vf.uni-lj.si; 2Veterinary Faculty, Institute of Pathology, Wild Animals, Fish and Bees, University of Ljubljana, Gerbičeva 60, 1000 Ljubljana, Slovenia; diana.zelevengust@vf.uni-lj.si

**Keywords:** small ruminant lentiviruses, SRLV, maedi-visna, caprine arthritis-encephalitis, chamois, mouflon, serology, ELISA, survey

## Abstract

**Simple Summary:**

Small ruminant lentiviruses (SRLVs) are responsible for the worldwide diseases maedi-visna and caprine arthritis-encephalitis in sheep and goats and are also widespread in Slovenian sheep and goats. The infection is lifelong with chronic inflammatory lesions in various organ systems. Cross-species transmission of SRLV strains in sheep and goats is well documented, but there are few data on the ability of these viruses to infect wild ruminants. The objective of this study was to investigate whether SRLVs circulate among wild small ruminants in Slovenia. Thirty-eight serum samples from free-ranging chamois and European mouflon were tested for antibodies against SRLV. A total of 1 mouflon tested seropositive, whereas all samples from chamois were negative.

**Abstract:**

Small ruminant lentiviruses (SRLVs) belong to the genus *Lentivirus* in the *Retroviridae* family, which are responsible for the diseases maedi-visna and caprine arthritis-encephalitis in sheep and goats worldwide and are also widespread in Slovenian sheep and goats. SRLVs cause lifelong infections with chronic inflammatory lesions in various organ systems. Cross-species transmission of SRLV strains in sheep and goats is well documented, but there are few data on the ability of these viruses to infect wild ruminants. The objective of this study was to investigate whether SRLVs circulate among wild small ruminants in Slovenia. During the 2017–2018 hunting season, a total of 38 blood samples were collected from free-ranging chamois (*Rupicapra rupicapra*) and European mouflon (*Ovis ammon musimon*). The serum samples were tested for antibodies against SRLV by enzyme-linked immunosorbent assay (ELISA). The serological tests revealed that of all tested mouflons, 1 animal (11.1%) was seropositive, while all samples from chamois were negative. Based on the results of this study and considering the results of previous studies in which SRLV infections were detected in mouflons with low seroprevalence, it is very likely that the detected seropositive animal was an incidental spillover host for SRLV. Although no seropositive samples were found in chamois, we cannot speculate on whether chamois may not be a host for SRLV infection because of the small sample size and the disadvantages of the ELISA assay used when applied to samples from chamois.

## 1. Introduction

Small ruminant lentiviruses (SRLVs) belong to the *Lentivirus* genus in the *Retroviridae* family, which are responsible for maedi-visna (MVV) and caprine arthritis-encephalitis (CAEV) diseases in sheep and goats worldwide [1] and are also widespread in Slovenian sheep and goats. Infection with SRLV affects various organ systems, including the central nervous system, lungs, joints, and mammary gland, where chronic inflammatory lesions occur. The incubation period can last months to years, and the infection is lifelong. The typical clinical manifestation in goats is chronic polyarthritis and in kids encephalitis, while in sheep, chronic pneumonia and mastitis are the predominant clinical manifestations [2,3,4,5]. Most infected animals show few clinical signs of disease but may transmit infection to others. Infections are mainly transmitted via colostrum from ewe to lamb and via the respiratory tract between animals in close contact [6,7]. After infection, SRLVs integrate into the host cell genome as a provirus, with the main target cells being monocyte-derived macrophages and dendritic cells [8].

There is no successful vaccine or treatment for SRLV, so the disease is controlled by identifying and eliminating infected animals and preventing new infections. SRLV infection is diagnosed by detecting specific antibodies against the virus using serological methods such as the agar gel immunodiffusion assay (AGID) and enzyme-linked immunosorbent assay (ELISA). ELISA is the most widely used screening test for the detection of specific antibodies against a variety of viral strains, and is generally a more sensitive technique than AGID. Several commercial ELISA tests are available. Several studies have also described the use of polymerase chain reaction (PCR) methods in blood and other tissues, with most studies using peripheral blood leukocytes as target cells for proviral genome detection [9,10].

Infections with lentiviruses of sheep and goats were previously considered species-specific, but molecular epidemiological studies in recent decades suggest that these viruses are capable of infecting both species [11,12,13]. While cross-species transmission of SRLV strains in sheep and goats is well documented, there are few data on the ability of these viruses to infect wild ruminants. Natural infections with CAEV have been detected in Alpine ibex (*Capra ibex*) [14] and Rocky Mountain goats (*Oreamnos americanus*) [15], while experimental infections with this virus have been reported in mouflon (*Ovis ammon musimon*) [16]. Serological evidence of SRLV infection has been reported in roe deer (*Capreolus capreolus*), red deer (*Cervus elaphus*), and mouflon [17,18].

In Slovenia, the estimated population size of the chamois is about 11,000 animals where it inhabits contiguous habitats in the Alps and Pre-alpine areas, and smaller distributions are found in the Dinaric region. The mouflon is a non-indigenous species in Slovenia, introduced in 1953, and its population size is estimated at 1500 animals. Its habitat is steep, sunny mountain slopes near the tree line [19]. In pastures where these species share their habitat with domestic ruminants, transmission of infectious diseases between these wild ruminants and domestic ruminants could occur. The results of previous studies of SRLV infections in wild ruminants suggest that wild ruminants may serve as virus reservoirs responsible for the infections of domestic small ruminants. Wild ruminants infected with SRLV can transmit the infection to domestic small ruminants upon contact during the grazing season. This could be a major obstacle to SRLV eradication programs in domestic sheep and goat herds [14].

The aim of this study was to investigate whether SRLVs circulate among wild small ruminants in Slovenia.

## 2. Materials and Methods

During the 2017/2018 hunting season (May 2017 to May 2018), blood samples were collected nationwide from a total of 38 apparently healthy adult wild small ruminants, including 29 chamois and 9 mouflons (Figure 1) of both sexes and different ages. The horn growth ring method was used for age determination. The age of the animals included in this study ranged from less than 1 year to more than 1 year, namely 1 to 18 years for chamois and 1 to 6 years for mouflon, respectively. The mean age of chamois was 2 years and that of mouflon was 1 year. Samples were collected by licensed game wardens and hunters from animals harvested during the regular annual cull. Because all samples were collected post-mortem, no ethical or animal welfare agency approval was required. Prior to sample collection, hunters were instructed on the procedure and provided with sample collection kits. Immediately after death, blood samples were collected from the jugular vein or heart. After collection, blood samples were transported to the Veterinary Faculty, University of Ljubljana, centrifuged at 1200× *g* for 10 min, and stored in the freezer at −20 °C until use.

Serum samples were analyzed for the detection of MVV/CAEV antibodies using a commercially available ELISA kit (Chekit-CAEV/MVV Screening ELISA Test Kit, IDEXX Laboratories) according to the manufacturer’s instructions. The results were interpreted according to the manufacturer’s instructions.

## 3. Results

The results of the ELISA test of the samples from chamois and mouflon are shown in Table 1. Serological examination revealed that of all tested mouflons, one animal (11.1%) was seropositive, namely a six-year-old male mouflon (OD value in ELISA test was 0.595 and calculated % value was 67%). All samples from chamois tested negative.

## 4. Discussion

In Slovenia, CAEV infection was first detected in a goat dairy herd in 1996, when severe chronic arthritis was diagnosed and specific antibodies for CAEV were detected. An initial serological survey in 1997 confirmed the presence of the disease in several sheep flocks [20]. In 2005, a comprehensive national surveillance program was implemented. Approximately 30,000 goats and sheep were tested for the presence of specific SRLV antibodies, revealing a seroprevalence of nearly 6%. A phylogenetic study of SRLV strains in sheep and goats in Slovenia revealed that Slovenian SRLV strains are highly heterogeneous, with sheep strains belonging to MVV (genotype A) and goat strains belonging to MVV (genotype A) and CAEV (genotype B), confirming the possibility of cross-species virus transmission [21]. The susceptibility of wild ruminants to SRLV has been previously demonstrated in roe deer, red deer, and alpine ibex in Europe [14,17,18] and also in Rocky Mountain goats [15]. To our knowledge, there are only two reports of SRLV antibody testing in mouflon, both from Spain [17,22], and one report from Italy [23] describing SRLV antibody testing in chamois.

This study is the first to investigate the exposure of wild small ruminants to SRLV in Slovenia. Samples were collected in hunting areas throughout the territory of Slovenia. The ELISA used in this study was validated for domestic sheep and goats. However, mouflons are closely related to sheep, being feral subspecies of domestic sheep, and chamois are a species from a different taxonomic genus than sheep (genus *Ovis*) and goats (genus *Capra*), namely the genus *Rupicapra*, but within the same subfamily *Caprinae* [24], so these ELISA tests should be able to detect antibodies to SRLV in mouflons and chamois. Regarding the serological diagnosis of SRLV infections in wild ruminants, some of the previously published studies have used commercial ELISA kits for domestic animals [15,16,22], while others have used in-house or adapted commercial ELISA kits [17,18].

The results of this study revealed one seropositive mouflon. Similar observations were made in Spain by Sanjosé et al. [17], who reported one seropositive sample out of 16 tested mouflon samples. On the other hand, the results of another study from Spain showed that none of the 101 wild mouflons tested had positive serological results against MVV [22]. Contact between free-ranging mouflon and domestic small ruminants is possible because these species share a habitat when sheep and goats are moved to pastures. In Slovenia, sheep and goats are moved to pastures in early spring and remain there until winter, so the seropositive animal could be infected by domestic small ruminants. In this study, no seropositive samples were detected in chamois. The fact that we did not detect antibodies against SRLV in chamois is consistent with a previous report from Italy [23], where no SRLV antibodies were detected in chamois. Diagnosis of infectious diseases in wild animals is more difficult than in domestic animals for several reasons. These include the difficulty of collecting samples from wild animals, poor quality of the samples, and lack of knowledge about the pathogenesis and epidemiology of wildlife diseases. In addition, ELISA tests for the serological diagnosis of infectious diseases are generally not validated for wildlife samples. It is not certain whether the diagnostic sensitivity or specificity of commercial diagnostic ELISA tests validated for domestic animals is different when applied to wildlife samples. The fact that we were unable to detect antibodies to SRLV in chamois could be due to a lack of infection in these species or due to the poor diagnostic sensitivity or specificity of ELISA tests when used for chamois samples. The small sample size and ELISA used in this study are a disadvantage in drawing a clear conclusion about the role of chamois as a host for SRLV infection. Further studies with larger sample sizes and using other diagnostic techniques or ELISA validated for wildlife species would be needed to draw such conclusions.

## 5. Conclusions

Due to the small sample size in this study, it is quite difficult to draw a definite conclusion about SRLV infections in mouflon. It is possible that SRLVs circulate in free-ranging mouflons in Slovenia, but considering the results of previous studies in which SRLV infections were detected in mouflons, it is very likely that the detected seropositive animal is an accidental spillover host for SRLV. Although no seropositive samples were found in chamois, we cannot speculate on whether chamois may not be a host for SRLV infection because of the small sample size and the disadvantages of the ELISA assay used when applied to samples from chamois.

## Figures and Tables

**Figure 1 animals-12-01032-f001:**
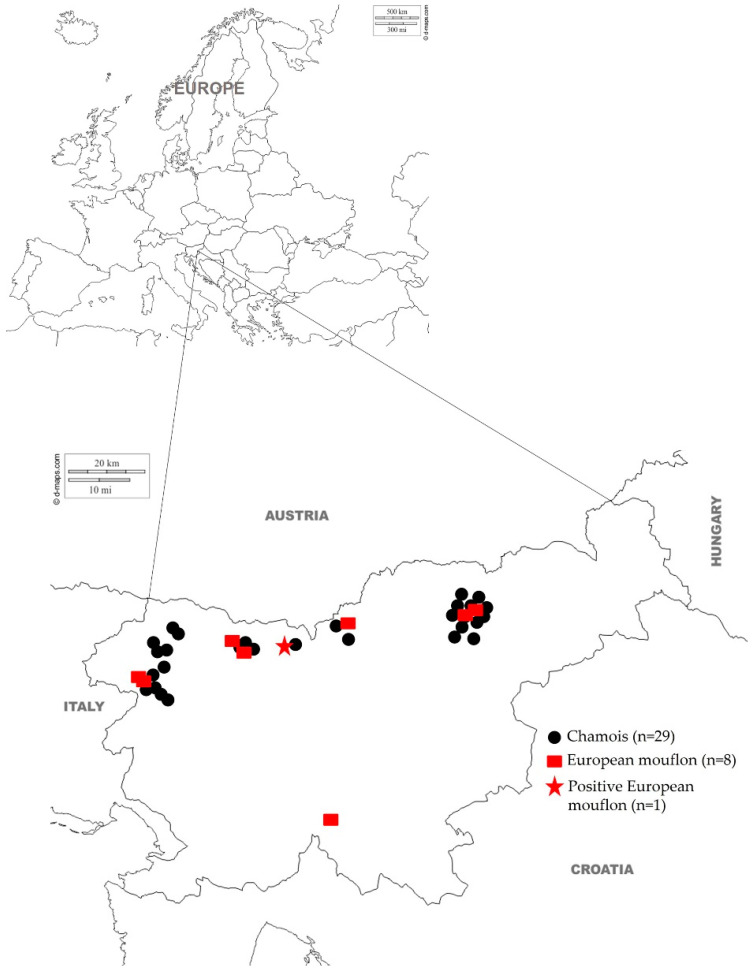
Geographical locations of SRLV antibody-negative and -positive samples from chamois and mouflon in Slovenia tested with ELISA.

**Table 1 animals-12-01032-t001:** Results of detection of specific antibodies against MVV/CAEV using ELISA in serum samples from wild small ruminants in Slovenia.

Species (No. and Sex; Mean Age)	ELISA	
	No. of Tested	No. (%) of Positive
Chamois (16 ♂, 13 ♀; 2) *	29	0
Mouflon (3 ♂, 6 ♀; 1) *	9	1 (11.1%)

* ♂ = male, ♀ = female.

## Data Availability

The data presented in this study are available on request from the corresponding author.

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
