# Peer review of "Serological Survey of Small Ruminant Lentivirus Infections in Free-Ranging Mouflon and Chamois in Slovenia"

_animals, 2022, doi:10.3390/ani12081032_

Round 1
Reviewer 1 Report
An interesting, well-written manuscript describing the serological survey of SRLV in mouflon and chamois in Slovenia.
It would be useful to include some information (if there is any) in the discussion about the supposed non-susceptibility or resistance to SRLV infection in wild ruminants (namely chamois and mouflon). Could there be an innate resistance to SRLV or could there be other reasons?
Probably it might be also useful to know the (presumed) current seroprevalence of SRLV in domestic small ruminants in Slovenia and whether the main hunting areas are close to pastures or grazing areas, where sheep and goats are kept during the summer season. Has direct contact between wild and domestic small ruminants been observed on these pastures?
P3, L101-104 and L106-109: Which genotype do you suspect as the cause of the seroconversion?
Author Response
FROM:
Gorazd Vengušt
Veterinary Faculty University of Ljubljana
Gerbičeva 60
SI-1000 Ljubljana
SLOVENIA
E-mail: gorazd.vengust@vf.uni-lj.si
Date: April 10th, 2022
TO:
Reviewer 1
Animals
Revised version of manuscript with ID animals-1645290 entitled: “Serological survey of small ruminant lentivirus infections in free-ranging mouflon and chamois in Slovenia “.
Authors: Urška Kuhar, Diana Žele Vengušt and Gorazd Vengušt
Dear Editor and Reviewer,
We would like to thank you for considering our manuscript for publishing in the Animals as well as for detailed review and useful suggestions that helped us to improve the quality of our manuscript. We have taken into consideration all of your comments and accepted most of your suggestions.
Concerning comments of the Reviewer1:
An interesting, well-written manuscript describing the serological survey of SRLV in mouflon and chamois in Slovenia.
It would be useful to include some information (if there is any) in the discussion about the supposed non-susceptibility or resistance to SRLV infection in wild ruminants (namely chamois and mouflon). Could there be an innate resistance to SRLV or could there be other reasons?
To the best of our knowledge there is no data regarding non-susceptibility or resistance to SRLV infection in wild small ruminants.
Probably it might be also useful to know the (presumed) current seroprevalence of SRLV in domestic small ruminants in Slovenia and whether the main hunting areas are close to pastures or grazing areas, where sheep and goats are kept during the summer season. Has direct contact between wild and domestic small ruminants been observed on these pastures?
There was a nationwide seroprevalence study in sheep and goats in 2005 revealing 6% seroprevalence (discussed in the manuscript). No systematic seroprevalence study was conducted later. In a frame of a PhD thesis there were samples collected in sheep and goat flocks on voluntary basis around 2010, 2011. Flocks were tested for antibodies detection, for virus detection with PCR and phylogenetic analysis was performed. But this study does not give a nationwide seroprevalence picture.
As suggested by another reviewer and the editor, all speculation about the epidemiological role of wild populations for SRLV infections was removed from the discussion and conclusions (also simple summary and abstract) because the sample size in this study was limited, preventing clear conclusions in this regard.
Discussion regarding when sheep and goat flocks move to pastures and about possible contact with mouflon was added in Lines 165-168.
P3, L101-104 and L106-109: Which genotype do you suspect as the cause of the seroconversion?
Considering the cross-species transmission of SRLV strains in sheep and goats we cannot speculate about the genotype in the infected mouflon.
Finally, we hope that the manuscript suits all criteria required. Expecting your answer, I remain
Sincerely yours,
Gorazd Vengušt
Reviewer 2 Report
The manuscript by Kuhar et al investigate SRLVs infection by testing serum samples collected from free-ranging chamois and European mouflon using commercial ELISA kit. However, it's better to test these serum samples using another method, like PCR, to confirm the results. Also, please include the raw data of ELISA.
Author Response
FROM:
Gorazd Vengušt
Veterinary Faculty University of Ljubljana
Gerbičeva 60
SI-1000 Ljubljana
SLOVENIA
E-mail: gorazd.vengust@vf.uni-lj.si
Date: April 10th, 2022
TO:
Reviewer 2
Animals
Revised version of manuscript with ID animals-1645290 entitled: “Serological survey of small ruminant lentivirus infections in free-ranging mouflon and chamois in Slovenia “.
Authors: Urška Kuhar, Diana Žele Vengušt and Gorazd Vengušt
Dear Editor and Reviewer,
We would like to thank you for considering our manuscript for publishing in the Animals as well as for detailed review and useful suggestions that helped us to improve the quality of our manuscript. We have taken into consideration all of your comments and accepted most of your suggestions.
Concerning comments of the Reviewer2:
The manuscript by Kuhar et al investigate SRLVs infection by testing serum samples collected from free-ranging chamois and European mouflon using commercial ELISA kit. However, it's better to test these serum samples using another method, like PCR, to confirm the results. Also, please include the raw data of ELISA.
The PCR for detection of virus nucleic acid can be performed on whole blood or tissue samples but not on serum samples.
The ELISA OD value and calculated % value for the positive sample was added to results (OD was 0.595 and calculated % value was 67%) in Line 124.
Finally, we hope that the manuscript suits all criteria required. Expecting your answer, I remain
Sincerely yours,
Gorazd Vengušt
Reviewer 3 Report
This communication is the first case of reporting serological evidences of SRLVs in wildlife in Slovenia and the second in Europe after the Spanish research mentioned in the text. Although sampling, on a national scale, is quite limited, this kind of report certainly deserves attention and should be stimulated.
The sampling of 29 chamois and 9 mouflon during a single hunting season in Slovenia is certainly a limited sampling and, as the authors rightly discussed, ELISA test for serological diagnosis of infectious diseases are generally not validated for wildlife species. However, it is to be welcomed that there are initiatives in this direction which, with preliminary data, can stimulate other studies in a systematic manner and based on significant sampling in relation to the target populations.
However, sampling to detect antibody response, in such a limited sample of animals, cannot assess whether there is a role of reservoir of the wild population compared to the domestic one, nor to exclude it in case of negative results, as stated in the simple summary. I suggest to limit the assessment to the sole relevance of the antibody reaction as a sign of pathogen circulation in the wild population and not to attempt a definition of the epidemiological role, nor of the interaction between wild and domestic sympatric population.
In this regard, I would like to point out that there is no information in the manuscript about the direct or indirect interactions of chamois or mouflon populations with the domestic small ruminant population. I suggest that some details should be given about the sympatric domestic population and, briefly, to how and at what time these flocks of sheep and goats are brought to pasture in order to provide effective indications of a possible interaction.
Specific comments
- Line 15 on Simple Summary. I suggest to use the term "wild" small ruminant to identify Chamois and Mouflon as from the epidemiological point of view also domestic sheep and goats, as they are managed on the Pastures, could be in fact free ranging (also lines 17, 27, 83, 88, 135, 153, etc.);
- Line 104 on Materials and Methods. I suggest to insert details related to the sympatric domestic population.
- Line 169. I suggest to specify that Chamois have not shown an antibody reaction, and this may suggest that they are not hosts, although further studies with other diagnostic techniques or with ELISA tests validated for wild species would be necessary to state no host role for chamois in SRLV infection.
Author Response
FROM:
Gorazd Vengušt
Veterinary Faculty University of Ljubljana
Gerbičeva 60
SI-1000 Ljubljana
SLOVENIA
E-mail: gorazd.vengust@vf.uni-lj.si
Date: April 10th, 2022
TO:
Reviewer 3
Animals
Revised version of manuscript with ID animals-1645290 entitled: “Serological survey of small ruminant lentivirus infections in free-ranging mouflon and chamois in Slovenia “.
Authors: Urška Kuhar, Diana Žele Vengušt and Gorazd Vengušt
Dear Editor and Reviewer,
We would like to thank you for considering our manuscript for publishing in the Animals as well as for detailed review and useful suggestions that helped us to improve the quality of our manuscript. We have taken into consideration all of your comments and accepted most of your suggestions.
Concerning comments of the Reviewer3:
This communication is the first case of reporting serological evidences of SRLVs in wildlife in Slovenia and the second in Europe after the Spanish research mentioned in the text. Although sampling, on a national scale, is quite limited, this kind of report certainly deserves attention and should be stimulated.
The sampling of 29 chamois and 9 mouflon during a single hunting season in Slovenia is certainly a limited sampling and, as the authors rightly discussed, ELISA test for serological diagnosis of infectious diseases are generally not validated for wildlife species. However, it is to be welcomed that there are initiatives in this direction which, with preliminary data, can stimulate other studies in a systematic manner and based on significant sampling in relation to the target populations.
However, sampling to detect antibody response, in such a limited sample of animals, cannot assess whether there is a role of reservoir of the wild population compared to the domestic one, nor to exclude it in case of negative results, as stated in the simple summary. I suggest to limit the assessment to the sole relevance of the antibody reaction as a sign of pathogen circulation in the wild population and not to attempt a definition of the epidemiological role, nor of the interaction between wild and domestic sympatric population. In this regard, I would like to point out that there is no information in the manuscript about the direct or indirect interactions of chamois or mouflon populations with the domestic small ruminant population. I suggest that some details should be given about the sympatric domestic population and, briefly, to how and at what time these flocks of sheep and goats are brought to pasture in order to provide effective indications of a possible interaction.
As also suggested by the editor, all speculation about the epidemiological role of wild populations for SRLV infections has been removed from the discussion in the conclusions (also simple summary and abstract ). Lines 16, 19, 20, 27, 35-39, 100, 101, 178-181, 187-191.
Discussion regarding when sheep and goat flocks move to pastures was added in Lines 165-168.
Specific comments
- Line 15 on Simple Summary. I suggest to use the term “wild” small ruminant to identify Chamois and Mouflon as from the epidemiological point of view also domestic sheep and goats, as they are managed on the Pastures, could be in fact free ranging (also lines 17, 27, 83, 88, 135, 153, etc.);
Corrected along the manuscript instead of free-ranging small ruminants, the term wild small ruminants was used instead. When the term free-ranging mouflon or chamois was used, then there is no doubt about the meaning, so this term was left as it is.
- Line 104 on Materials and Methods. I suggest to insert details related to the sympatric domestic population.
The population of domestic small ruminants was not a subject of this study.
- Line 169. I suggest to specify that Chamois have not shown an antibody reaction, and this may suggest that they are not hosts, although further studies with other diagnostic techniques or with ELISA tests validated for wild species would be necessary to state no host role for chamois in SRLV infection.
Text as suggested was added in Lines 36-39, 178-181, 189-191.
Finally, we hope that the manuscript suits all criteria required. Expecting your answer, I remain
Sincerely yours,
Gorazd Vengušt
Round 2
Reviewer 2 Report
1. Line 56, change ". and" to ", and";
2. Table 1, Mouflon (3 males +7 females=10), but the No. of tested is 9. Please clarify it.
Author Response
FROM:
Gorazd Vengušt
Veterinary Faculty University of Ljubljana
Gerbičeva 60
SI-1000 Ljubljana
SLOVENIA
E-mail: gorazd.vengust@vf.uni-lj.si
Date: April 11th, 2022
TO:
Reviewer 2
Animals
Revised version of manuscript with ID animals-1645290 entitled: “Serological survey of small ruminant lentivirus infections in free-ranging mouflon and chamois in Slovenia “.
Authors: Urška Kuhar, Diana Žele Vengušt and Gorazd Vengušt
Dear Editor and Reviewer,
We have taken into consideration your comments and accepted your suggestions.
Concerning comments of the Reviewer2:
- Line 56, change ". and" to ", and";
Corrected.
- Table 1, Mouflon (3 males +7 females=10), but the No. of tested is 9. Please clarify it.
A typing mistake in the Table. Corrected in the table to 6 females.
Finally, we hope that the manuscript suits all criteria required. Expecting your answer, I remain
Sincerely yours,
Gorazd Vengušt